# Captive Bolt Gun-Related Vascular Injury: A Single Center Experience

**DOI:** 10.3390/diagnostics14100977

**Published:** 2024-05-08

**Authors:** Jure Pešak, Andrej Porčnik, Borut Prestor

**Affiliations:** Department of Neurosurgery, University Medical Centre Ljubljana, 1000 Ljubljana, Slovenia; andrej.porcnik@kclj.si (A.P.); borut.prestor@kclj.si (B.P.)

**Keywords:** captive bolt gun, vascular injury, low-velocity penetrating brain injury, skin–bone imprimatum, incomplete bone fragment debridement, frontopolar artery territory, anterior cranial fossa bolt trajectory

## Abstract

This article investigates the clinical and radiological characteristics of captive bolt gun head injuries, a rare form of low-velocity penetrating brain injury. Eleven consecutive patients were included in the study. Vascular injuries and the rate of infection were systematically analyzed. Radiological findings reveal common bolt trajectories in the anterior cranial fossa, with identified risk factors for a poor outcome including trajectory crossing midline, hematocephalus, and paranasal sinus involvement. Only one patient had a good outcome. Despite meticulous microsurgical techniques, this study highlights often unfavorable clinical outcomes in captive bolt gun injuries, with vascular injury identified as a potential contributing risk factor for a poor outcome. Knowledge of variant vascular tree anatomy and corresponding vascular territory is important. To avoid potential vascular injuries, a complete removal of bone fragments was not always performed and it did not increase the rate of infection, challenging the conventional wisdom advocating for the complete removal of bone fragments. These findings contribute novel insights into captive bolt gun-related injuries, paving the way for further research.

## 1. Introduction

Captive bolt gun (CBG) head injuries represent a rare form of penetrating brain injuries with only a few documented cases in the literature [1,2,3,4,5,6,7]. This is a device that is commonly employed in the meat industry to stun animals before slaughter [1,2]. A bolt penetrates the scull, causing direct and indirect injures to the brain parenchyma through bone fragments [1,2]. The bolt length varies among different tool brands, ranging from 60 to 90 mm in various series [1,3,4]. CBG head injuries are classified as low-velocity penetrating brain injuries, given that the bolt’s peak velocity ranges between 30 and 60 m per second [1,3,4]. In the rural areas of Middle and Eastern European countries, where the GBG is the most prevalent stun device in the meat industry, it can be utilized in suicide attempts [2,4]. CBG injuries resulting from suicide attempts are usually anticipated in the anterior and middle cranial fossa. The trajectory depth does generally not exceed the bolt length, as the punched-out skin and bone fragments do not act as a secondary projectile due to the low velocity [1]. Moreover, there are typically no exit wounds. Despite being considered low-velocity brain injuries, these patients often have unfavorable outcome [1]. The reason may be due to the additional vascular injuries that are caused directly by bolt or bone fragments. Vascular injuries can result in ischemic lesion in the corresponding vascular territories. These lesions are highly variable, especially in the anterior cerebral artery territory [8,9]. Variations are greater in the smaller branches of the vascular tree [8]. The anatomic distribution of vascular territories can be radiologically assesed despite anatomical variations in the vascular tree [8,9]. There are three known radiological factors of a poor outcome in penetrating head injury: trajectory crossing midline, hematocephalus, and vascular injury [2,7].

The first case studies of CBG head injury were published in 1960 [10]. Sparse clinical case reports, mostly featuring 1–3 case studies per paper, shed light on early clinical findings of the pathology [5,10,11,12]. Following these initial 13 cases on PubMed written in the English language, the first miniseries of 12 clinical cases was published in 2002, augmented by supplementary experimental data in the field of forensic medicine [4]. The ongoing period of forensic medicine research has advanced to the most developed stage in CBG head injury research to date, expanding the knowledge of specific mechanisms of CBG head injuries through experiments and reports with complicated cases of CBG head injuries [13,14,15,16,17,18,19,20]. Our clinical study presents a single-center experience with CBG head injuries. Potential clinical and radiological prognostic factors were analyzed. It is the first report where CBG-related vascular injuries have been systematically analyzed.

## 2. Materials and Methods

The clinical study was conducted at the Department of Neurosurgery of the University Medical Centre Ljubljana, Slovenia. All patients with CBG head injury who received either conservative or operative neurosurgical care were included. From December 2016 to December 2023, eleven consecutive patients with CBG head injuries were studied. Medical records were retrospectively analyzed for clinical and radiological factors. Additional telephone interviews with patient relatives were conducted on 13 December 2023, for further clarification of treatment outcomes. General patient data and the time of the incident were recorded. Initial neurological status was evaluated based on the Glasgow coma scale (GCS) at the scene of the incident and on the GCS with pupillary status at the time of admission. Radiological factors of CBG head injury were evaluated using the initial computed tomography (CT) scan at admission. Localization of the entry wound, trajectory length, localization of parenchymal damage, paranasal sinus involvement, hematocephalus occurrence, and midline crossing were recorded. Treatment strategy was tailored according to the clinical status at admission and to the initial CT scan. The surgical technique was noted, considering the degree of bone fragment removal and the effectiveness of paranasal sinus reconstruction. Postoperative ischemic lesions were evaluated within the first 24 h using a CT scan. The radiological signs of secondary ischemic lesion, infection-related injuries, and shunt-dependent hydrocephalus were evaluated by utilizing diagnostic imaging during the postoperative period. The clinical outcome was evaluated based on the Glasgow outcome score (GOS). A favorable outcome was determined as a GOS of 4 or 5, an unfavorable outcome was determined as a GOS of 2 or 3, and death was determined as a GOS of 1. Data are presented along with descriptive statistics.

## 3. Results

### 3.1. Patient Characteristics

All of the 11 patients were men (Table 1). The mean age was 59 ± 12 years (with an interval of 36–94 years). Six patients had a psychiatric history and two patients had major comorbidities—carcinoma. While seven patients had an initial Glasgow coma scale (GCS) score of 14 or above at the scene, only three patients maintained a GCS score of 14 or higher at the time of hospital admission. Three patients presented with a GCS of 3 at the scene and with dilated and unresponsive pupils at admission. The time from the injury to hospital admission could not be reliably established. The mean GCS score at hospital admission was 7 ± 5.

### 3.2. Radiological Characteristics

The entry wounds were located in the pterional region in five patients, in the parasagittal region in five patients, and in the temporal region in one patient (Figure 1). The mean penetrating trajectory length was 80 ± 15 mm (an interval of 48–95 mm). The brand of the CBG could not be determined. All patients exhibited cortico-subcortical parenchymal injury while seven patients had corpus callosum involvement and four patients displayed involvement in the deep structures or in the brainstem. Nine patients developed intracerebral hemorrhage.

Three patients had paranasal sinuses involvement, eight patients developed hematocephalus, and the CBG trajectory crossed midline in five patients. Two patients had one proposed radiological factor for a poor outcome while six patients had two factors for a poor outcome and a single patient had all three factors for a poor outcome (Table 1).

The head computed tomography model depicts entry bone wounds marked by circles and corresponding trajectories marked by lines within the circles for all patients. Nine injuries were on the right side and two were on the left side of the head. Trajectories only show their approximate direction and not their depth. For the ease of visualization, all injuries on the left side were symmetrically translocated to the right side. Surgically treated cases are denoted by red circles. Cases treated with only palliative conservative care are represented by black circles and black trajectories. Two of the surgically treated patients with ischemic areas due to the vascular injury of the anterior circulation are presented by green trajectories. Red trajectories indicate surgically treated cases without major vascular injury. The blue trajectory shows the case with the injury of the superior sagittal sinus.

### 3.3. Primary Treatment and Surgical Technique

Out of the eleven patients, seven underwent surgical treatment (64%). The treatment was performed within 4 h from hospital admission. In cases involving paranasal sinuses, a cranialization of the frontal sinus was performed in all surgical cases, accompanied by additional reconstruction of the galea aponeurotica. A craniotomy was the primary surgical approach in six cases while craniectomy was performed in only one case. Full debridement of intraparenchymal bone fragments was achieved in only two patients, considering the potential for additional iatrogenic brain or vascular injury, prompting caution for complete removal in other cases. Four patients in poor initial condition at admission received only palliative conservative care. Indications for conservative palliative care included a poor clinical condition with symptoms of brain death and/or irreversible neural injuries incompatible with a functional outcome based on the initial head CT scan. All patients who underwent surgical treatment received prophylactic antibiotic treatment for six weeks.

### 3.4. Radiological and Clinical Outcome

Four patients developed diffuse brain edema without brain perfusion in the first 24 h—all of whom were under palliative conservative care. In those four patients, vascular injury could not be determined. Two out of seven surgically treated patients developed incomplete ischemic lesion in a named small branching artery territory in anterior circulation. Both lesions were in the vascular territory of the frontopolar artery (Figure 2). Notably, five surgically treated patients did not develop an ischemic lesion, as outlined in Table 1.

One patient developed an intracranial abscess with secondary infection-related parenchymal injury necessitating secondary surgery. One patient developed shunt-dependent hydrocephalus.

The computed tomographic angiography of patient 10 reveals a typical captive bolt gun trajectory in the anterior cranial fossa. Blue-marked bone fragments caused significant direct injury to the brain parenchyma and the left frontopolar artery, as indicated by the purple arrow highlighting the branching stem of the artery with an absence of distal blood flow. Bone fragments located under the circle of Willis pose a potential risk to both the anterior and posterior groups of the perforators. Arteries susceptible to injury include the supraclinoid carotid artery segments, the left middle cerebral artery, and the anterior cerebral artery (marked in red), along with the contralateral middle cerebral artery perforators (the parent artery marked in orange). Injury of the vertebrobasilar system (marked in green) occurs when the trajectories are deeper.

## 4. Discussion

Head injuries with CBG represent a rare form of low-velocity penetrating brain injury characterized by specific clinical and radiological features. All of our case studies were instances of self-inflicted harm and all were men as described by other researchers [4]; therefore, all injuries were found in the anterior and middle cranial fossa and mostly on the right side. The age range varied significantly, from 36 to 94 years. More than half of the patients exhibited diagnosed depression, which is consistent with the literature [1,2,4,5], and two patients had significant comorbidities.

Prompt intervention in patients with good initial neurological status may spare a substantial portion of brain parenchyma and its vasculature from a secondary injury. However, an initial GCS score of 3 at the scene with unresponsive and dilated pupils at admission significantly correlates with poor outcomes [2]. In our series, despite the absence of three proposed radiological risk factors and of a postoperative ischemic lesion, one surgically treated patient with good initial performance status succumbed to injury, emphasizing the impact of serious comorbidities in the very elderly (a 94-year-old man). A larger series reported mortality rates between 60 and 90% [4,5,6]. Despite CBG head injuries being categorized as low-velocity penetrating brain injuries, a high mortality is expected due to the self-inflicted mechanism, the high probability of vascular injury, and the high rate of infection.

Similar to the findings in the literature, common entry points were in the right pterional and frontobasal midline areas [4]. Projections of bolt trajectories often include vascular structures. One of the three proposed radiological factors for a poor outcome includes vascular injury such as injury of superior sagittal sinus and its bridging veins and/or anterior cerebral artery branches in the midline, which were injured in two cases in our series. With a mean trajectory length of 80 mm, bolt and/or bone fragments usually do not cause significant direct injury to the brainstem but often reach the anterior and posterior groups of the perforators (Table 1, Figure 2) [8]. Gnjidić et al. reported that the cause of death in five out of seven successful suicides was an injury to blood vessels in the Sylvian fissure and in one out of seven successful suicides, the injury was on the internal carotid artery [4].

The complete removal of all bone fragments with meticulous debridement is advocated to reduce the risk of intracranial infection [4,6]. However, contrary to these authors, our series experienced incomplete debridement of intraparenchymal bone fragments in all but one surgical case, without late postoperative intracranial infection. An intracranial abscess in one case that necessitated secondary surgery, despite complete removal of all bone fragments during the primary surgery, was likely due to latent cerebrospinal fluid leakage through a paranasal sinus fistula. Shunt-dependent hydrocephalus developed in this case because severe ventriculitis occurred. Also, this was a case where craniotomy was performed. In our series, only one out of seven surgically treated cases involved craniectomy. Described are cases with either craniotomy or craniectomy with mixed results [2,4,5,6,7]. We consistently employed galea reconstruction and sinus cranialization to separate cranial spaces in cases of paranasal sinus communication, mitigating cerebrospinal fluid-related ascending infections. Complete removal of bone fragments does not seem crucial, especially if iatrogenic injury to the perforators or small branching arteries could occur. Due to the colonization of skin flora at the end of the trajectory in their series, the importance of prolonged antibiotic prophylactic treatment in our surgical cases is emphasized [4]. Geisenberger et al. further showed that almost all cases had skin–bone imprimatum that can explain cutaneous bacterial flora [1]. In our series, the only case requiring secondary surgery due to cerebral abscess had mixed bacterial flora, including non-skin-related Enterococcus casseliflavus and Enterococcus durans. Unusual microbes causing infection might explain this complication leading to secondary surgery.

Even though there was no delayed cerebral ischemia in any named artery’s territory or in the anterior or posterior perforator group territory, according to vascular territories described by Vogels et al. [8], two of our surgically treated patients developed an ischemic lesion in the frontopolar artery territory. In both cases, this led to an unfavorable clinical outcome (Table 1).

## 5. Conclusions

Captive bolt gun head injuries present as low-velocity penetrating brain injuries. This study is the first to shed light from the perspective of vascular injury. Despite a comprehensive understanding of variant anatomy, mechanism of injury, microsurgical techniques, appropriate antibiotic treatment, and intensive care, these injuries often result in unfavorable clinical outcomes. The type of surgical treatment depends on the entry wound location, paranasal sinus involvement, and bolt trajectory. Our study indicates that incomplete removal of deep bone fragments and craniotomy do not elevate infection rates, necessitating secondary surgery. This is important because additional iatrogenic vascular injury due to fragment removal can lead to a worse outcome.

## Figures and Tables

**Figure 1 diagnostics-14-00977-f001:**
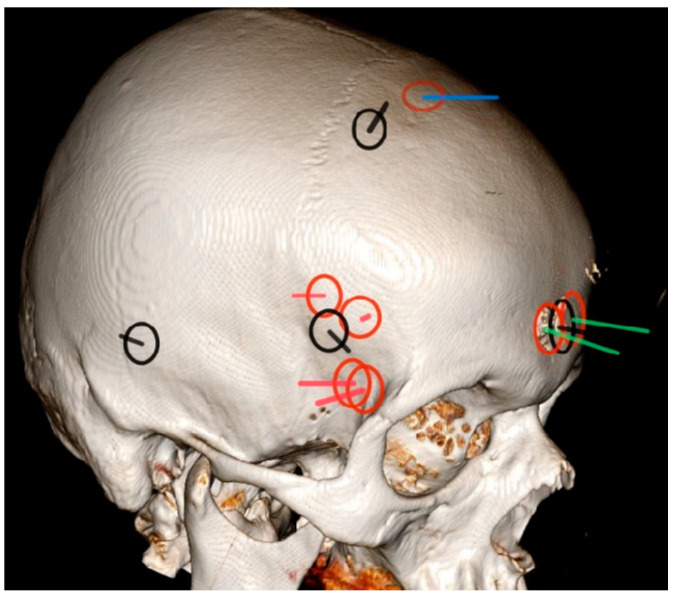
Schematic representation of the points of bone entry wound location and trajectory course.

**Figure 2 diagnostics-14-00977-f002:**
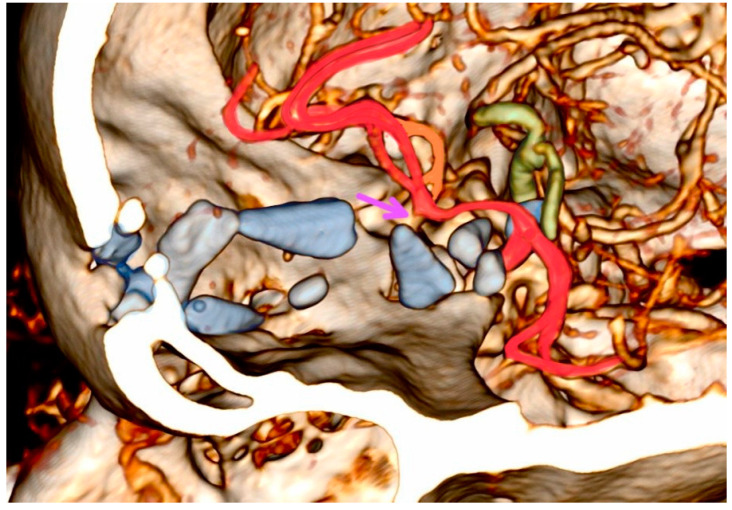
Three-dimensional head computed tomographic angiography reconstruction of patient 10.

**Table 1 diagnostics-14-00977-t001:** Clinical and radiological characteristics of patients.

Patient	1	2	3	4	5	6	7	8	9	10	11
Gender	male	male	male	male	male	male	male	male	male	male	male
Age	58	65	68	36	94	67	65	43	40	52	67
GCS at scene	15	15	11	3 *	15	14	10	15	3 *	3 *	14
GCS at admission	3	15	3	3	15	3 *	6	15	3 *	3 *	3
Latest GOS	3	3	3	1 *	1	1 *	3	5	1 *	1 *	3
Comorbidities	bipolar	depression	depression	depression	carcinoma	none	none	carcinoma, depression	none	none	depression
PBI entry wound location and side	parasagittal (R)	pterional (R)	pterional (R)	parasagittal (R)	parasagittal (L)	parasagittal (R)	pterional (R)	pterional (R)	temporal (R)	pterional (R)	parasagittal (L)
Trajectory length	82 mm	83 mm	90 mm	82 mm	48 mm	86 mm	83 mm	68 mm	93 mm	95 mm	81 mm
Parenchymal damage	CS + CC	CS	CS + CC	CS + CC	CS	CS + CC + deep structures	CS	CS + CC	CS + deep structures	CS + deep structures	CS + CC + deep structures
Sinuses involvement	frontal	no	no	no	no	frontal	ethmoidal	no	no	no	frontal
Hematocephalus	yes	no	yes	yes	no	yes	no	yes	yes	yes	yes
Midline crossing	no	no	yes	yes	no	no	no	no	yes	yes	yes
Secondary ischemic lesion	frontopolar territory	no	no	no	no	no	no	no	no	no	frontopolar territory
Secondary infection	yes	no	no	x	x	x	no	no	x	x	no
Shunt-dependent hydrocephalus	yes	no	no	x	x	x	no	no	x	x	no

* Treatment stage when the patient was categorized for palliative care—conservative symptomatic treatment. GCS—Glasgow coma scale. GOS—Glasgow outcome scale. PBI—penetrating brain injury. R—right side. L—left side. CS—cortico-subcortical. CC—corpus callosum. x—Not applicable because the patient died before secondary endpoints could be measured.

## Data Availability

The raw data supporting the conclusions of this article will be made available by the authors on request.

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
