# Peer review of "Captive Bolt Gun-Related Vascular Injury: A Single Center Experience"

_diagnostics, 2024, doi:10.3390/diagnostics14100977_

Round 1

Reviewer 1 Report

Comments and Suggestions for Authors

This article investigates the clinical and radiological characteristics of captive bolt gun head injuries, a rare form of low-velocity penetrating brain injury. The risk factors for adverse clinical outcomes are analyzed, including radiological features and vascular injuries.

Here are some issues in the article that need to be addressed:

1.The patient data analysis employes descriptive statistical methods, which are relatively subjective and do not provide insight into how the conclusions are reached. Additionally, the number of patients included is small, leading to significant heterogeneity.

2.The author's analysis do not consider some factors that may affect patient prognosis, such as the time from injury to discovery, timely treatment, surgical intervention, the model of the captive bolt gun, past medical history, and others.

3.Tables 1 and 2 do not clearly present the injury information for each patient, only providing an overall description of the entire population, making it difficult for readers to understand the correlation between different radiological features and patient prognosis. Combining the localization of entry wound, trajectory length, localization of parenchymal damage, paranasal sinus involvement, occurrence of hematocephalus, midline crossing, secondary ischemic lesion, infection-related injuries, and shunt-dependent hydrocephalus into a single table for clearer presentation might be a good choice.

4.In Figure 1, please avoid symmetrically transferring all injuries from the left side to the right side, as it does not facilitate observation. Additionally, provide a clearer description of the ballistic trajectories in the figure. Are different lengths of line segments indicative of different trajectory lengths? If so, consider adding a scale to the figure for better assessment of trajectory length. Do different angles of the line segments represent different entry angles? These aspects are not clearly explained in the text.

5.Some conclusions lack corresponding data and descriptions in the text. For example, the systematic analysis of vascular injuries and infection rates, as well as the identification of risk factors such as psychiatric history, male gender, employment in the meat industry, rural environment, and low education level, for captive bolt gun injuries, require more detailed explanation and supporting data.

Comments on the Quality of English Language

The quality of English language usage is good, but minor revisions are recommended for enhanced clarity and fluidity.

Author Response

1.Summary:

Thank you very much for taking the time to review this manuscript and found some points where we can improve our manuscript. Please find the detailed responses below and the corresponding revisions in the re-submitted files.

2.Questions for general evaluations

Most of the answers to your comment are included in the section point-by-point response.

Comments and Suggestions for Authors

This article investigates the clinical and radiological characteristics of captive bolt gun head injuries, a rare form of low-velocity penetrating brain injury. The risk factors for adverse clinical outcomes are analysed, including radiological features and vascular injuries.

Here are some issues in the article that need to be addressed:

  1. 3. Point-by-point response to Comments and Suggestions for Authors

Comment 1: The patient data analysis employes descriptive statistical methods, which are relatively subjective and do not provide insight into how the conclusions are reached. Additionally, the number of patients included is small, leading to significant heterogeneity.

Response 1: Considering the small number, as you pointed out, the statistical power is of course weak. This is the reason for descriptive statistical method. We must take into consideration the rarity of the topic. Moreover the the number of cases in other articles are also very low-as is also written in our article (page 2, line 41-52).

Comment 2:The author's analysis do not consider some factors that may affect patient prognosis, such as the time from injury to discovery, timely treatment, surgical intervention, the model of the captive bolt gun, past medical history, and others.

Response 2: This is also one of the limitation of this report. The time form injury to discovery was mostly not known, so we could not analyse this. (Added  in line 87)Timely treatment form hospital admission was mostly the same. Patients were operated in 4 hours after the hospital admission (added line 130-131), model of captive bolt gun could not be determined (added line 106). We added past medical history (table 1 and line 83)

Comment 3.Tables 1 and 2 do not clearly present the injury information for each patient, only providing an overall description of the entire population, making it difficult for readers to understand the correlation between different radiological features and patient prognosis. Combining the localization of entry wound, trajectory length, localization of parenchymal damage, paranasal sinus involvement, occurrence of hematocephalus, midline crossing, secondary ischemic lesion, infection-related injuries, and shunt-dependent hydrocephalus into a single table for clearer presentation might be a good choice.

Response 3: We agree, thank you for this advise. We have joined both tables into one with added medical comorbidities including psychiatric history. The new table can be found between line 92-101.

Comment 4.In Figure 1, please avoid symmetrically transferring all injuries from the left side to the right side, as it does not facilitate observation. Additionally, provide a clearer description of the ballistic trajectories in the figure. Are different lengths of line segments indicative of different trajectory lengths? If so, consider adding a scale to the figure for better assessment of trajectory length. Do different angles of the line segments represent different entry angles? These aspects are not clearly explained in the text.

Response 4: Thank you for clarifying this Figure. Since most of the injuries were on the right side (9/11) we decided to have just one figure – creating the other for the left side in our opinion is not necessary, and might add samo additional disturbance for the readers. But as you had suggested we have written this explanation in the text. Unfortunately, our study is clinical and not forensic, so therefore trajectories shown on the Image are just approximations and are based on trajectories present on head CT. They are meant just to give the reader an idea where the bolt went. We have added this explanation in the figure text 117-127.  Different lengths of line segments do not mean different trajectory lengths- we have added this explanation in the figure text as well 119-130.  It would be hard to make exact trajectories length and angles for all patients in one picture. For these a study for each individual patient would be needed in three planes, which would be interesting though, but this was not the aim of our study.

Comment 5: Some conclusions lack corresponding data and descriptions in the text. For example, the systematic analysis of vascular injuries and infection rates, as well as the identification of risk factors such as psychiatric history, male gender, employment in the meat industry, rural environment, and low education level, for captive bolt gun injuries, require more detailed explanation and supporting data.

Response 5:

You are absolutely right. The conclusions about employment in the meat industry, low education level in and rural environment were removed form our conclusions. This demographic data are also not the aim of our series. We added data about the infection rates and vascular injuries- Line 141-142, line 146-147. Some changes were made also in conclusion (line 172-174) and abstract.

Comments on the Quality of English Language

The quality of English language usage is good, but minor revisions are recommended for enhanced clarity and fluidity.

Response: We did minor revisions in English language.

Thank you once again for reviewing the paper. I hope we managed to answer and clarify all your comments.

Reviewer 2 Report

Comments and Suggestions for Authors

This is a very well written and comprehensive analysis of a series of 11 patients with low velocity penetrating head trauma. This provide insight into uncommon brain injury be captive bolt guns and emphasises the resulting vascular injury and subsequent ischaemia as the main reason for the predominant poor outcome. Tha analysis provided is excellent and the accompanying figures are most informative. 

The highlight is identifying the factors associated with poor outcome (although these are based on just 11 cases) and the low infection sequelae despite non clearance of all bone fragments perhaps mitigated by long period of prophylactic antibiotics. 

Author Response

1.Summary:

Thank you very much for taking the time to review this manuscript. Please find the detailed responses below and the corresponding revisions in the re-submitted files.

Reviewers comment:

This is a very well written and comprehensive analysis of a series of 11 patients with low velocity penetrating head trauma. This provide insight into uncommon brain injury be captive bolt guns and emphasises the resulting vascular injury and subsequent ischaemia as the main reason for the predominant poor outcome. Tha analysis provided is excellent and the accompanying figures are most informative. 

The highlight is identifying the factors associated with poor outcome (although these are based on just 11 cases) and the low infection sequelae despite non clearance of all bone fragments perhaps mitigated by long period of prophylactic antibiotics. 

 Response1:

Thank you very much for your nice review and pointing out the most important aspect of our article.

Round 2

Reviewer 1 Report

Comments and Suggestions for Authors

 The article has been improved through revisions, but there are still a few minor problems remaining, as follows:

1.In Table 1, there seems to be an issue with the place marked with "**" in the row for 'parenchymal damage'. Do the 'x' and 'no' in the rows for 'secondary infection' and 'shunt-dependent hydrocephalus' mean the same thing? If so, please unify them.

2.On line 124, '(table 2)' needs to be removed. Alternatively, you can keep the original Table 2, which is a descriptive statistical summary for all patients.

Comments on the Quality of English Language

The quality of English language usage is good.

Author Response

Summary:

Thank you for taking the time to review this manuscript and suggesting further improvements. Please find the detailed responses below and the corresponding revisions/corrections highlighted/in track changes in the re-submitted files.

Answers to your comment are included in the section point-by-point response:

Comment 1: In Table 1, there seems to be an issue with the place marked with "**" in the row for “parenchymal damage”. Do the “x” and “no” in the rows for “secondary infection” and “shunt-dependent hydrocephalus” mean the same thing? If so, please unify them.

Response 1: Due to uncertainties, we have completed the table under “parenchymal damage” and clarify the meaning of “X”. X does not have the same meaning as “no”.

Comment 2: On line 124, “(table 2)” needs to be removed. Alternatively, you can keep the original Table 2, which is a descriptive statistical summary for all patients.

Response 2: We agree that “(table 2)” should be removed.

Thank you once again for reviewing the paper. I hope we managed to answer and clarify all your comments.
